# Purification, Identification and Evaluation of Antioxidant Peptides from Pea Protein Hydrolysates

**DOI:** 10.3390/molecules28072952

**Published:** 2023-03-25

**Authors:** Dan Zhao, Xiaolan Liu

**Affiliations:** 1Key Laboratory of Food Science and Engineering of Heilongjiang Province, College of Food Engineering, Harbin University of Commerce, Harbin 150028, China; 2Key Laboratory of Corn Deep Processing Theory and Technology of Heilongjiang Province, College of Food and Bioengineering, Qiqihar University, Qiqihar 161006, China

**Keywords:** purification, antioxidant activity, molecular docking, molecular mechanism

## Abstract

Food-derived antioxidant peptides can be explored as natural antioxidants due to their potential health benefits. In this study, antioxidant peptides were isolated and purified from pea protein hydrolysates (PPH). The DPPH and ABTS radical scavenging activities were used as indexes to purify the antioxidant peptides by a series of purification steps including ultrafiltration, ion exchange chromatography, G25 gel filtration chromatography, and reversed-phase chromatography. Three novel antioxidant peptides YLVN, EEHLCFR and TFY were identified, which all exhibited strong antioxidant activity in vitro. EEHLCFR showed stronger DPPH scavenging activity with an IC_50_ value of 0.027 mg/mL. YLVN showed stronger ABTS scavenging activity with an IC_50_ value of 0.002 mg/mL and higher ORAC values of 1.120 ± 0.231 μmol TE/μmol, which is even better than that of GSH. Three novel antioxidant peptides significantly elevated LO2 cells viability even at the concentration of 0.025 mg/mL, and cell viability enhanced to 53.42 ± 1.19%, 55.78 ± 1.03%, and 51.09 ± 1.06% respectively, compared to that of H_2_O_2_ injury group (48.35 ± 0.96%), and prevented the accumulation of ROS by enhancing the activities of antioxidant enzymes in H_2_O_2_-induced oxidative stress LO2 cells. The molecular docking results showed that the potential molecular mechanism of the three novel antioxidant peptides may be in high correlation with the activation of the Keap1-Nrf2 pathway by occupying the Keap1-Nrf2 binding site. These results demonstrate that the three novel antioxidant peptides are potential natural antioxidants that can be devoted to medicine or functional food ingredients.

## 1. Introduction

Under normal physiological conditions, reactive oxygen species (ROS) production and elimination are in a dynamic equilibrium, mainly because the organism has a complex antioxidant defense system, including enzymatic and non-enzymatic [1]. However, under pathological conditions, the amount of ROS could exceed the defense capacity of the antioxidant system leading to oxidative stress [2]. Oxidative stress can damage cell membranes and biological molecules; sustained oxidative stress damage could consequently give rise to a variety of chronic diseases like cardiovascular diseases, diabetes, obesity, and neurodegenerative diseases [3], while it could be alleviated by the provision of exogenous antioxidants [4].

The use of various natural and synthetic antioxidants has been reported to protect the body from oxidative damage and various diseases. Compared to the potential harm of synthetic antioxidants, natural antioxidants from food sources have attracted extensive attention due to their wide range of sources and better safety. Protein hydrolysates from plant sources such as hemp seed [5], lotus seed [6], soybean [7], hazelnut [8] and corn [9], as well as from animal sources such as casein [10], Chinese mutton ham [11], Antarctic Krill [12], swim bladders [13] and sea cucumber [14] all have shown an excellent capacity for scavenging free radicals like ROS, inhibiting lipid peroxidation and protein oxidation.

Peas are rich in starch and protein, low in fat, and contain many important vitamins and minerals [15]. Because of its high starch content, the pea is mainly used to produce pea vermicelli. Pea protein content ranges between 20 and 30% of the total content, rich in glutamic acid, lysine, and leucine, which is a good source of essential amino acids. As a by-product of pea starch production, the development and utilization of pea protein is not sufficient.

Our previous study showed that pea protein hydrolysates (PPHs) by alcalase protease had better antioxidant activity in vitro. However, information about the structural and activity characteristics of antioxidant peptides from pea protein is limited, and the underlying mechanism of antioxidant capacity has not been revealed. Therefore, the objective of this study was to (1) isolate, purify, identify and characterize the peptides with strong antioxidant activities; (2) evaluate the antioxidant activities and cytoprotective effect in vitro and (3) do preliminary research on the antioxidant mechanism through molecular docking. Our findings of this study would be available for utilization of pea protein in functional foods as antioxidants.

## 2. Results and Discussion

### 2.1. Purification of Pea Antioxidative Peptides 

#### Subsubsection

Firstly, PPH was fractionated by 3 kDa and 5 kDa ultrafiltration membrane, and three MW fractions, PPH-I (permeate < 3 kDa), PPH-II (3–5 kDa) and PPH-III (retentate > 5 kDa) were obtained. There was no significant difference in the DPPH and ABTS free radical scavenging value among the three fractions. Since low molecular weight peptides are easier to absorb, PPH-I was taken for further purification. The activities (DPPH and ABTS) of fractions from each purification step were tracked, and the flow chart of the entire separation and purification stage is shown in Figure 1.

Pea protein hydrolysate by alcalase protease was a mixture of peptides with different charges, and PPH-I fractions were purified by Q Sepharose FF. As shown in Figure 2A, the purification yielded two major fractions, and most peptides of PPH-I were negatively charged under pH7.5 conditions, so they could be concentrated and purified by Q Sepharose FF. In two fractions, bound fraction II exhibited high DPPH and ABTS radical scavenging activities (Figure 2B), and was selected for Sephadex G-25 column purification. Three major peaks of fractions II-1 II-2 and II-3 were eluted from the G25 gel filtration column (Figure 3A), and fraction II-2 was the major active fraction with a DPPH and ABTS radical scavenging value of 45.53% and 36.32% respectively, and showed a molecular weight distribution of about 307–1421 Da (Figure 3B).

The reversed-phase column Pronto SIL C18 was used for further separation and purification of Fraction II-2. As shown in Figure 4A, many antioxidant fractions eluted from the Pronto SIL C18 column, fraction 6 (named FII-2-6) exhibited the highest DPPH and ABTS radical scavenging values of 39.22% and 32.74% respectively, which was significantly higher than that of other 14 fractions (*p* < 0.05) (Figure 4B). Therefore FII-2-6 was selected for further purification and analysis. The FII-2-6 was further purified by an analytical column named Cosmosil pbr (Ø4.6 × 250 mm, 5 μm), and the results are shown in Figure 4C. Nineteen chromatographic peaks were obtained, and all of them exhibited active values to a certain extent, of which peaks 3, 5, and 7 (named FII-2-6-3, FII-2-6-5, FII-2-6-7) showed a higher active value of DPPH radical scavenging activities of 33.93%, 34.23% and 32.73%, respectively, and ABTS radical scavenging activities of 28.53%, 32.91 and 30.20% respectively (Figure 4D). The Xselect TM CSH130 (Ø4.6 × 250 mm, 3.5μm) column was selected for further separation of fractions of FII-2-6-3, FII-2-6-5 and FII-2-6-7, respectively. As shown in Figure 5A–C, each of these three components showed a single peak by secondary reverse phase, which was collected respectively for determining the amino acid sequences.

### 2.2. Sequence Identification and Antioxidant Activities of Three Antioxidant Peptides

The sequence of peptides was identified based on LC-MS/MS, and the results are shown in Figure 6A–C and molecular mass and amino acid sequence of three peptides are shown in Table 1. The mass spectrum of Figure 6A illustrated the FII-2-6-3 was identified as Tyr-Leu-Val-Asn (YLVN, named Peptide 1) with MW of 507.59 Da, correspondingly Figure 6B illustrated FII-2-6-5 was Glu-Glu-His-Leu-Cys-Phe-Arg (EEHLCFR, named Peptide 2) with MW of 933.06 Da, and Figure 6C illustrated FII-2-6-7 was Thr-Phe-Tyr (TFY, named Peptide 3) with MW of 429.47 Da. The sequence of the peptide was searched in the BIOPEP-UWM database but no corresponding bioactive peptide sequences were retrieved. Therefore, it can be concluded that the peptides obtained in this study are three novel peptides.

The identified peptides were synthesized by the FMOC method and their antioxidant capabilities were investigated by DPPH·, OH·, ABTS^+^, O_2_^−^· and ORAC assays, and GSH was used as the positive control. The IC_50_ values of the three antioxidant peptides are shown in Table 2. When tested in the concentration range of 0.001 to 3.00 mg/mL, Peptide 2 exhibited the strongest DPPH scavenging ability among the three antioxidant peptides, its IC_50_ value of 0.027 ± 0.303 mg/mL was significantly lower than the GSH IC_50_ value of 0.081 ± 0.175 mg/mL (*p* < 0.05). In addition, it also has high OH· and O_2_^−^·radical scavenging activities. Moreover, the DPPH IC_50_ values of Peptide 3 were less than those of antioxidant peptides from the Antarctic Krill ESP9-ESP15 [16], lotus seed (LSPH:2.9 mg/mL) [6], sesame protein hydrolysate (RDRHQKIG: 4.648 ± 0.021 mg/mL, TDRHQKLR: 6.353 ± 0.035 mg/mL, MNDRVNQGE: 6.763 ± 0.084 mg/mL, RENIDKPSRA: 3.650 ± 0.117 mg/mL) [17].

All three novel peptides have exhibited strong ABTS free radical scavenging abilities. The IC_50_ values of Peptide 1 are significantly lower than that of GSH (*p* < 0.05), and the IC_50_ value of Peptide 3 is comparable to that of GSH. The ABTS+ radical scavenging capacities of Peptide 2 were a little weaker than that of Peptide 1 and 3, but also lower than that of jackfruit seed (VGPWQK: 1.0 mg/mL) [18], Chinese dry-cured mutton ham (MHP: 0.76± 0.12 mg/mL) [11], Chinese chestnut (VYTE: 0.13 ± 0.01, VSAFLA: 0.28 ± 0.04 mg/mL) [19].

The ORAC value results showed that all three peptides have high ORAC values, Peptide 1 and Peptide 2 exhibited the highest ORAC activities (1.120 ± 0.231 µmol TE/mg and 0.921 ± 0.131 µmol TE/µmol, respectively) even higher than that of GSH (0.484 ± 0.190 µmol TE/µmol) (*p* < 0.05). A higher ORAC value indicates a stronger ability of the peptides to stop the chain reactions of peroxyl radicals. These results indicated that the isolated peptides from pea protein hydrolysate are excellent antioxidant compounds with strong free radical scavenging activity [20].

### 2.3. Relationship of Peptides Structure-Antioxidant Activity

In general, short-chain peptides are considered to be more easily absorbed and more likely to have interactions with target free radicals. As shown in Table 1, short peptides with three to seven amino acid residues were identified in this study and with molecular weights of 507.59, 933.06 and 429.47 Da. In addition, in this work, the ratio of the hydrophobic amino acid in Peptide 1 YLVN, Peptide 2 EELHCFR and Peptide 3 TFY were 50%, 28.57% and 33.33% respectively, which exert antioxidant properties by providing H+ and enhancing synergy with other amino acids. This is consistent with the conclusions of previous studies that peptides which are rich in hydrophobic amino acids might be an important contributor to antioxidant activity e.g., peptide AAEYPA and peptide AKPGVY which contained 66.67% and 50% of hydrophobic amino acids, respectively, showed significant contribution to the radical-scavenging activities [21]. The peptides TSSSLNMAVRGGLTR and STTVGLGISMRSASVR contained 46.66% and 50% of hydrophobic amino acids and possessed high DPPH radical scavenging activity [22].

Both Peptide 1 (YLVN) and Peptide 3 (TFY) have tyrosine residues in their sequences. ABTS radicals can react with any hydroxylated aromatic compound, thus, whether the aromatic amino acid Tyr sequence is located at the C-terminus or N-terminus of the peptide, the phenolic -OH group in Tyr can act as a free radical scavenger by directly supplying hydrogen to the ABTS radical, making ABTS a stable molecule [23,24,25]. Similarly, the peptide ATVY purified from black sharkskin protein hydrolysate showed high ABTS radical scavenging activities due to the Tyr at the C-terminal [25]. The ORAC method is based on the mechanism of HAT, and the phenolic and indole groups of Tyr and Trp have the ability of hydrogen donors, so peptides containing these amino acids show high ORAC values [26]. Both Peptide 1 and Peptide 3 have high ORAC values, especially peptide 1 having a higher ORAC value than that of GSH, possibly due to the contribution of Tyr residues.

Polar amino acids including acidic and basic amino acids such as Glu and Arg were of great importance for peptides with antioxidant activities, due to the carboxyl and amino groups in their side chains which can enhance antioxidation by preventing the oxidation–reduction reaction of metal ions [27]. Peptide 2 (EEHLCFR) contains two Glu residues and one Arg residue which might contribute to its antioxidant activity. Notably, Peptide 2 EEHLCFR containing repeated amino acids Glu-Glu (EE) exhibit free radical reduction activity. This was consistent with the results of Wen [28], who observed that the amino acid residues of peptides RDPEER and KELEEK with repeating units had better DPPH and ABTS scavenging abilities. In addition, Peptide 2 EEHLCFR showed stronger OH- and O_2_^−^ scavenging activity compared to other peptides, due to the existence of Cys residue. The thiol group of cysteine could donate an electron to reactive oxygen radicals, by stabilizing the free radicals and showing antioxidant activity [17]. In a study, two peptides SYPTECRCR and SYPTECRMR were designed to confirm the effect of Cys residue on the antioxidant activity of the peptide. The experimental results found the antioxidant activity of SYPTECRCR is stronger than that of SYPTECRMR, so it could be concluded that Cys residue contributes to the antioxidant activity of peptides [17]. Furthermore, previous studies have found His (H), mainly present at the N-terminus or middle of antioxidant peptides, also has OH^−^ and O_2_^−^ radical quenching activity due to its imidazole ring [28], therefore, Cys and His would make a significant contribution to the antioxidant activities of EEHLCFR when they are present in the same peptide chain. In summary, the inference of the relationship between the structure and activity of the three identified peptides gives us reason to believe that the three novel peptides could exert excellent antioxidant capacity.

### 2.4. Cytoprotective Effect of Three Novel Antioxidant Peptides on H_2_O_2_ Damaged LO2 Cells

First, the experimental concentration intervals that do not have a particularly significant toxic effect on the cells were determined. As shown in Figure 7A the three peptides had no cytotoxicity to LO2 cells after treatment for 24 h at concentrations below 1 mg/mL. H_2_O_2_ is an important reactive oxygen species with relatively stable properties. It is often used as a model drug for oxidative stress damage in vitro due to less environmental pollution and less toxic effect on the human body. As shown in Figure 7B, when the concentration of H_2_O_2_ increased, the cell viability in the model group decreased. Previous studies have shown that cell viability of 50% is suitable as the optimal condition for H_2_O_2_-induced damage cell models. Therefore, the H_2_O_2_ concentration of 4 mmol/L and stimulation LO2 cells 2 h were selected for the damage model.

As shown in Figure 7C, compared to the H_2_O_2_-damaged model group, preincubation with Peptide 1 (YLVN), Peptide 2 (EEHLCFR), and Peptide 3 (TFY) significantly increased the viability of the H_2_O_2_ damaged LO2 cells. Notably, when the concentration of the three peptides was 0.025 mg/mL, the cell viabilities of YLVN, EEHLCFR, and TFY treatment groups were 53.42 ±1.19%, 55.78 ± 1.03%, and 51.09 ± 1.06%, respectively, indicating a better protective effect on H_2_O_2_ damaged LO2 cells than that of GSH positive control group (50.00 ± 0.39%). The results showed that the three novel antioxidant peptides present significant cytoprotective effects against the H_2_O_2_-induced oxidative damage in LO2 cells. Similarly, Hong [29] found that GSQ antioxidant peptides identified from Chinese leek seeds exhibited a stronger cytoprotective effect against H_2_O_2_-injured LO2 cells in a dose-dependent manner.

### 2.5. Effects of Three Antioxidant Peptides on the Levels of ROS, SOD, GSH-Px and CAT Activities in H_2_O_2_ Injured LO2 Cells

It is generally acknowledged that oxidative stress is caused by an imbalance between the oxidation and antioxidant system, resulting in the accumulation of intracellular ROS. oxidative stress can enter a vicious cycle because the ROS produced can destroy biomolecules, which leads to higher ROS accumulation [30]. As shown in Figure 8A, the fluorescence intensity of the H_2_O_2_ injured group was 214.16 ± 4.16%, which was significantly higher than that of the blank control group and peptides treatment group (*p* < 0.01), indicating that, a large amount of ROS was produced in the cell after H_2_O_2_ treatment. However, the fluorescence intensity was significantly reduced after peptides were pretreated even at a peptide concentration of 0.025 mg/mL. These results revealed that three novel peptides can significantly reduce ROS levels in cells by H_2_O_2_-induced oxidative stress, thereby alleviating oxidative damage.

GSH-Px, CAT, and SOD are important components of the enzyme antioxidant system. To further clarify whether the ability of three antioxidant peptides to protect H_2_O_2_-stressed LO2 cells from oxidative damage is related to the activation of endogenous antioxidant enzymes (SOD, CAT, GSH-PX), the activity of SODCAT, GSH-PX were measured. As shown in Figure 8B–D, after exposure to 4 mM H_2_O_2_ for 2 h, the H_2_O_2_ treatment group without pre-incubation with three peptides, the activities of GSH-Px, CAT and SOD decreased by 68.15%, 61.51% and 56.62%, respectively, versus the control group. While after pretreatment with three novel peptides at 0.01–0.1 mg/mL, the enzyme activity of H_2_O_2_-stressed cells increased in a dose-dependent manner.

GSH-Px is a class of antioxidant enzymes widely distributed in the body. Almost all organic hydroperoxides (ROOHs) can be reduced to ROH under the action of GSH-Px [31]. According to Figure 8B, GSH-Px activity decreased from 165.68 ± 0.48 U/mg (control group) to 52.75 ± 0.26 U/mg (H_2_O_2_ group). Three antioxidant peptide pretreatments significantly increased the GSH-Px activity of damaged cells. Notable, the activity of GSH-Px was the highest (165.48 ± 1.48 U/mg) and occurred in the EEHLCFR pretreatment group at a concentration of 0.1 mg/mL, which was close to the level of the blank control group.

CAT is widely present in cells and tissues and acts as a catalyst for the breakdown of H_2_O_2_, thereby avoiding oxidative damage in the body. As shown in Figure 7C, the CAT level in the damaged group was reduced from 16.68 U/mg to 6.42 ± 1.12 U/mg, compared with the control group. The three antioxidant peptide intervention groups significantly increased the activity of CAT, even at a concentration of 0.01 mg/mL, the CAT activity of peptides 1–3 pretreated groups increased by 33.95%, 69.15%, and 72.58%, respectively.

The main function of SOD is to clear intracellular O_2_^−^·and produce non-toxic H_2_O_2_. According to Figure 8D, Peptide 1(YLVN) presented a more obvious impact on increasing the activity of SOD than that of the other two peptides. Compared with the H_2_O_2_-treated group, the activities of SOD in LO2 cells pre-incubated with Peptide 1 from 0.01–0.1 mg/mL increased up to 1.5-, 1.58-, 1.67- and 1.77-fold, respectively. In an earlier study, researchers found that hemp seed protein hydrolysates (HPH) at a concentration of 0.4 mg/mL enhanced the activities of SOD (2.6-fold), CAT (2.5-fold) and GSH-Px (2.1-fold) in H_2_O_2_-treated HepG2 cells, compared with the H_2_O_2_-treated group [32].

These results indicate that Peptide 1 YLVN, Peptide 2 EEHLCFR and Peptide3 TFY could protect LO2 cells from H_2_O_2_-induced oxidative stress damage by inhibiting ROS production and activating endogenous antioxidant defense systems including GSH-Px, CAT and SOD, thereby maintaining the balance between intracellular and non-enzymatic to inhibit H_2_O_2_-induced cell oxidative damage.

### 2.6. Molecular Docking of Three Antioxidant Peptides with Keap1

The Keap1-Nrf2 pathway is one of the important defense pathways of cytoprotective responses to obliterate oxidative stress [33]. In this pathway, the Keap1 protein acts as a negative regulator of Nrf2 and specifically binds to the Nrf2 protein and eventually leads to the ubiquitination and degradation of Nrf2 by the Cul3 protein. Therefore, if peptides could occupy the binding site of Keap1 and Nrf2, it can improve the ubiquitination level of Nrf2, thereby improving the body’s antioxidant capacity [34]. Whether three peptides, YLVN, EEHLCFR and TFY, have the ability to inhibit Keap1-Nrf2 interactions was investigated in this study, by molecular docking, and the possible mechanism of interference with Keap1-NRF2 interaction at the molecular level was elucidated.

As shown in Table 3, the affinities of YLVN, EEHLCFR and TFY to Keap1 protein were −8.2, −7.2, and −8.9 kcal/mol, respectively, and TFY showed strong binding to keap1 than the other peptides.

The -Nrf2-bound domain can be divided into five subregions: P1, P2, P3, P4 and P5 [34]. Figure 9 shows the 3D and 2D interactions between Keap1 (4IQK) and three antioxidant peptides.

YLVN can form hydrogen bonds and carbon-hydrogen bonds with Arg415 in the P1 pocket, Asn382 and Asn414 in the P2 pocket, and Ser602 in the P3 pocket, and form pi-alkyl interactions with Tyr525 and Tyr572 in the P4 pocket. EEHLCFR can form hydrogen bonds with Arg415 and Arg483 in the P1 pocket, Arg380 and Asn414 in the P2 pocket, Ser555 in the P3 pocket, Gln530 in the P4 pocket, and form pi-alkyl groups with Phe577 in the P5 pocket and Ala556 in the P3 pocket, also interacting to form hydrophobic binding with Tyr572 in the P4 pocket. TFY can form hydrogen bonds with Arg415, and Ser508 in the P1 pocket, Ser363 in the P2 pocket, and hydrophobic binding with Tyr334 in the P5 pocket. These amino acid residues are essential amino acids for the binding of ETGE motifs and DLG motifs to Keap1. Therefore, the three peptides bind to amino acid residues in these pockets through hydrogen bonding and hydrophobic forces, forming a steric hindrance, thereby inhibiting the binding between Nrf2 and Keap1 [35]. Combined results of cell experiments, we speculated that YLVN, EEHLCFR, and TFY can enter the interior of the Kelch domain pocket, forming hydrogen bonds and hydrophobic forces with multiple hydrophobic amino acids in it, forming a steric hindrance, thereby inhibiting the binding between Nrf2 and Keap1. The Keap1-Nrf2-ARE signaling pathway was activated, and ultimately, the activities of SOD, CAT and GSH-Px increased.

## 3. Materials and Methods

### 3.1. Materials and Chemicals

Pea protein was supplied by Shandong Jian-yuan Co., Ltd. (Jinan, China). Protease Alcalase (2.35 × 10^4^ U/mL) was purchased from Novo Nordisk (Bagsvaerd, Denmark). Q Sepharose FF was purchased from GE Healthcare (Boston, MA, USA). Sephadex G-25 was purchased from Ruida Henghui Co., Ltd. (Beijing, China). Pronto SIL C18 and XselectTM CSH130 columns were purchased from Unimicro (Shanghai, China) and Waters Technologies Co., Ltd. (Milford, MA, USA), respectively. Cosmosil pbr column was purchased from Suzhou Mike Wangzhi Biotechnology Co., Ltd. (Suzhou, China). LO2 cells were purchased from Nanjing KGI Biotechnology Co., Ltd. (Nanjing, China). All other chemicals are of analytical reagent grade and bought from Shenggong Biotech Co., Ltd. (Shanghai, China).

### 3.2. Preparation of Pea Protein Hydrolysate

Pea proteins were dissolved in deionized water at a substrate concentration of 7 g/100 mL, and pH was adjusted to 8.5 using 1.0 mol/L NaOH, followed by the addition of 3.8% (enzyme/substrate ratio) alcalase protease to the system. After hydrolysis at 50 °C for 3 h, the mixture was boiled at 100 °C for 15 min to terminate the reaction. Afterwards, the enzymatic hydrolysate was centrifuged to obtain pea protein hydrolysates (PPH), which were freeze-dried for subsequent analysis.

### 3.3. Isolation and Purification of Antioxidant Peptides

The PPH was purified using 3 kDa and 5 kDa ultrafiltration membrane, and the fractions of >5 kDa, 3 kDa–5 kDa, and <3 kDa were collected. The fraction of the smaller than 3 kDa was preliminarily purified by the Q Sepharose Fast Flow column. The samples were dissolved in 20 mmol/L Tris-HCL buffer with 1 mol/L NaCl (pH 7.5), then put on a column (1.6 × 25 cm) of Q-Sepharose Fast Flow equilibrated with the same buffer, and eluted for 4 column volumes with a linear gradient of NaCl (0–1 M) at a flow rate of 2 mL/min and monitored at 214 nm. DPPH and ABTS radical scavenging activities of each fraction were measured and the fractions with the strongest scavenging activities were pooled and lyophilized. The lyophilized fractions were dissolved in distilled water at a concentration of 10 mg/mL and purified by Sephadex G-25 chromatography with a flow rate of 1 mL/min and monitored at 214 nm. The fractions with the highest DPPH and ABTS radical scavenging activities were pooled and lyophilized (molecular weight ranges from 307–1412 Da).

The lyophilized fractions were dissolved in 2% acetonitrile containing 0.065% TFA at a concentration of 10.0 mg/mL and loaded onto the ProntoSIL C18 (10 × 250 mm, 10 μm) column. A solution of 2% acetonitrile containing 0.065% TFA was used as buffer A; 80% acetonitrile containing 0.05% TFA was used as buffer B. A linear gradient elution program was used: 0–10 min, 100% buffer A; 10.1–60 min, 0–100% buffer B; 60–65 min, 100% buffer B; 65–70 min, 100–0% buffer B; injection volume: 500 μL; elution flow rate: 1 mL/min.

The fraction with the highest DPPH and ABTS antioxidant activities was further purified using an RP-column named Cosmosil pbr (4.6 × 250 mm, 5 μm). A solution of 2% acetonitrile containing 0.065% TFA was used as buffer A; 80% acetonitrile containing 0.05% TFA was used as buffer B. A linear gradient elution program was used: 0–5 min, 100% buffer A; 5.1–50 min, 0–40% buffer B; 50–60 min, 40–0% buffer B; injection volume: 10 μL; elution flow rate: 1 mL/min. The fractions with the highest DPPH and ABTS antioxidant activities were further separated and purified using XselectTM CSH130 column (4.6 × 150 mm, 3.5 μm) with the same elution conditions of Cosmosil pbr. The highest active fractions were collected, lyophilized and prepared to identify the sequence of the purified peptide.

### 3.4. Identification of Pea Antioxidant Peptides

The sequences of purified pea antioxidant peptides were determined by mass spectrogram in Beijing Batar Parker Biotechnology Co., Ltd. (Q Exactive™ Hybrid Quadrupole-Orbitrap™ Mass Spectrometer) (Beijing, China).

### 3.5. Synthesis of Peptides

Three purified peptides were synthesized by Qiang Yao Biotechnology Co., Ltd. (Wuhan, China) with a final purity ≥ 98% verified by HPLC (Gilson L7420, Hitachi Limited, Tokyo, Japan).

### 3.6. Antioxidant Activity of Three Novel Antioxidant Peptides In Vitro

#### 3.6.1. DPPH Radical Scavenging Activity

DPPH radical scavenging activity was determined using the method described by Pownall et al. [36]. Briefly, 100 μL of peptide or glutathione samples solution at different concentrations (0.01~3.0 mg/mL) and 100 μL of DPPH solution were mixed in a 96-well microplate. After being incubated at room temperature for 30 min in the dark, the absorbance at 517 nm of the resulting solution was measured (A_sample_) and methanol was used as control (A_control_). The percentage of DPPH radical scavenging activity was calculated as [(A_control_ − A_sample_)/A_control_] × 100.

#### 3.6.2. Hydroxyl Radical (OH·) Scavenging Activity

The OH· scavenging assay was determined using the method described by Pownall et al. [36]. Briefly, 50 μL of peptide or glutathione sample solution of different concentrations (0.01~3.0 mg/mL) were first added to a 96-well microplate followed by the addition of 50 μL of 5 mmol/L 1,10-phenanthroline (in phosphate buffer), 50 μL of 5 mmol/L FeSO4 (in water) and 50 μL of 1% H_2_O_2_ aqueous solution, and incubate the reaction mixture at 37 °C for 60 min, the absorbance at 510 nm of the supernatant was determined (A_sample_). The blank group replaced the hydrolysate and H_2_O_2_ with distilled water (A_blank_), and the control group replaced the hydrolysate with distilled water (A_control_). The percentage of OH· scavenging activity was calculated as [(A_sample_ − A_control_)/(A_blank_ − Ac_ontrol_)] × 100.

#### 3.6.3. ABTS Radical Scavenging Activity

ABTS radical scavenging activity was determined using the method described by Li et al. [37]. Preparations of ABTS stock solution: 7 mmol/L of ABTS reagent and 2.45 mmol/L of potassium persulfate were mixed in a volume of 1:1, and incubated at room temperature in the dark for 12~16 h to make the ABTS solution. Preparation of the ABTS working solution: Dilute the stock solution with 0.01 mol/L PBS so that the absorbance value of the solution at 734 nm is 0.70 ± 0.02.

Solutions of 100 μL of peptide or glutathione samples of different concentrations (0.001~1 mg/mL) and 100 μL of diluted ABTS+ working solution were added to the 96-well plate, incubated at room temperature for 15 min, and the absorbance value of the reaction system at 734 nm was determined (A_sample_). PBS replaced the sample solution as a control group (A_control_). The percentage of ABTS radical scavenging activity was calculated as [1 − (A_sample_/A_control_)].

#### 3.6.4. Superoxide (O_2_^−^) Scavenging Activity

The superoxide scavenging activity was determined using the method described by Pownall et al. [36]. Samples of 80 μL peptide or glutathione of different concentrations (0.05~3 mg/mL), Tris-HCl (50 mmol/L pH 8.3 containing 1 mmol/L EDTA) and 40 μL of catetriol with 1.5 mmol/L HCl (10 mmol/L) were mixed. The reaction rate (ΔA/min) was measured immediately at 320 nm every 1 min at room temperature for 5 min continuously, using the buffer as a control (A_control_). The percentage of superoxide scavenging activity was calculated as [(∆A_control_/min − ∆A_sample_/min)/∆A_control_/min] × 100.

#### 3.6.5. Oxygen Radical Absorption Capacity (ORAC)

The ORAC assay procedure followed the method of Nimalaratne et al. [38]. The analysis was carried out using black 96-well microplates, in which 20 μL of peptide or glutathione of different concentrations (0.001~3 mg/mL) was mixed with 120 μL of Fluorescein and incubated for 5 min at 37 °C in the microplate. Then, 60 μL of AAPH was added and the microplate was shaken for 2 min. The plate was immediately placed in the reader, and the fluorescence was recorded every 2 min for 180 min (emission and excitation wavelengths were 485 nm and 528 nm, respectively). PBS was used as a blank and Trolox solutions were used as positive controls. The ORAC values of samples were quantified as Trolox equivalent (TE) antioxidant capacity.

The scavenging activities of DPPH, OH, ABTS^+^, and O_2_^−^ were expressed as IC_50_ value (the half maximal inhibitory concentration). The ORAC value was expressed as µmol TE/μmol sample.

### 3.7. Effects of Peptides on H_2_O_2_-Induced Oxidative Stress in LO2 Cells

#### 3.7.1. Determination of Cell Viability

The cell viability of LO2 cells was measured using the MTT method, according to the method of Xia et al. [39] with slight modification. Briefly, LO2 cells were cultured with peptides and GSH concentrations from 0.005–1 mg/mL for 24 h, then MTT solution was added and samples were incubated for another 4 h, the absorbance at 490 nm was measured.

#### 3.7.2. Establishment of H_2_O_2_ Oxidative Damage Model and Protective Effect on LO2 Cells

LO2 cells were treated and cultured for 24 h, 100 μL of different concentrations of 0.5, 1, 2, 4, 6, and 8 mmol/L of H_2_O_2_ were added to each well, and 6 parallel wells were set up for each well. After culturing for 2 h, the cell viability was measured by the MTT method to determine the concentration of hydrogen peroxide damage.

The experiment settings were grouped as follows: the blank control group; the H_2_O_2_-induced group; the antioxidant peptides treatment group and GSH positive control group. LO2 cells were cultured normally for 24 h, 100 μL of culture medium was added to the blank control group and the H_2_O_2_-induced group; to the antioxidant peptides group 100 μL concentration of 0.025, 0.05, and 0.1 mg/mL peptides solution were added, and to GSH group 100 μL concentration of 0.025 mg/mL GSH solution was added and continued to cultivate for another 24 h. Then 100 μL of 4 mmol/L H_2_O_2_ solution was added to all groups except the blank control group, and incubated for 2 h. Cell viability was determined by MTT assay.

#### 3.7.3. Measurement of Intracellular ROS

LO2 cells were cultured according to the method of Section 3.7.2, and then ROS was analyzed according to the method of Xia [39] and Cermeño et al. [40]. Briefly, LO2 cells were seeded at a density of 1 × 10^5^ cells per well into black 96 well plates which were then incubated at 37 °C in 5% CO_2_ for 24 h. Cells were grouped according to the method in Section 3.7.2 above. After a 24 h culture, the culture medium was discarded and 100 μL of DCFH-DA working solution (10 μmol/L) was added to each well and incubated for 30 min. The supernatant was then discarded and cells were washed with PBS. The fluorescence intensity was then measured with a microplate reader (excitation at 485 nm and emission at 528 nm).

#### 3.7.4. Analysis of Antioxidant Enzyme Activities

Groups and cell cultures were prepared according to the method described in Section 3.7.2. The SOD, CAT and GSH-Px experiments were performed according to the Human glutathione peroxidase (SOD, CAT and GSH-Px) enzyme-linked ELISA kit instructions (Jianglai Biotechnology, China).

### 3.8. Molecular Docking Simulation

The interactions between the peptide and target protein were estimated by molecular docking. The 3D structure of the Keap1 protein (PBD ID: 4IQK) was downloaded from the PDB protein database (RCSB PDB: Homepage) as a receptor molecule. The structure of the three novel peptides was drawn by MarvinSketch software (chemaxon.com) (accessed on 16 January 2023) and optimized for energy minimization as a ligand. Molecular docking was performed using Autodock vina. The box size was 58 × 66 × 92, and the center position was X: −32.308, Y: −0.668, Z: 0.0. Finally, Pymol software was used to visualize the docking results.

### 3.9. Statistical Analysis

Analyses were performed in triplicate, and the results were expressed as mean ± standard deviation (SD, *n* = 3). The SPSS 26.0 software was used for one-way ANOVA where *p* < 0.01 indicates a very significant difference, *p* < 0.05 indicates a significant difference.

## 4. Conclusions

In this study, three novel antioxidant peptides YLVN, EEHLCFR, and TFY were purified from pea protein hydrolysates. All three peptides exhibited excellent antioxidant activities in vitro and by reducing ROS levels and upregulating GSH-Px, CAT and SOD activities to protect H_2_O_2_-damaged LO2 cells. The three novel antioxidant peptides reported in this work can affect the interaction of Keap1-Nrf2 by occupying the binding site of the Keap1–Kelch domain by molecular docking test. In addition to the activation of the Nrf2-Keap1 signaling pathway proposed in this study, there are a variety of signaling pathways involved in the regulation of cellular oxidative stress, which needs to be further studied in conjunction with complex in vivo experiments. The above results suggest that YLVN, EEHLCFR, and TFY have the potential to be used as a source of natural antioxidants.

## Figures and Tables

**Figure 1 molecules-28-02952-f001:**
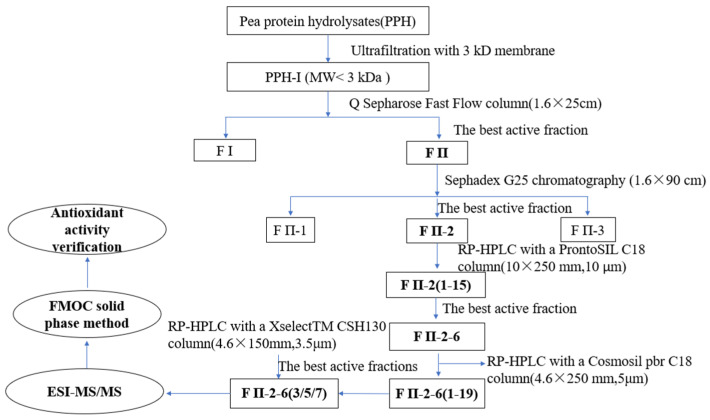
The flow chart of purifying antioxidant peptides from pea proteins hydrolysates. F II, F II-2, F II-2-6, and F II-2-6(3/5/7) are the active peak components determined by the Q Sepharose FF, Sephadex G-25, Pronto SIL C18 column, and Cosmosil pbr column, respectively.

**Figure 2 molecules-28-02952-f002:**
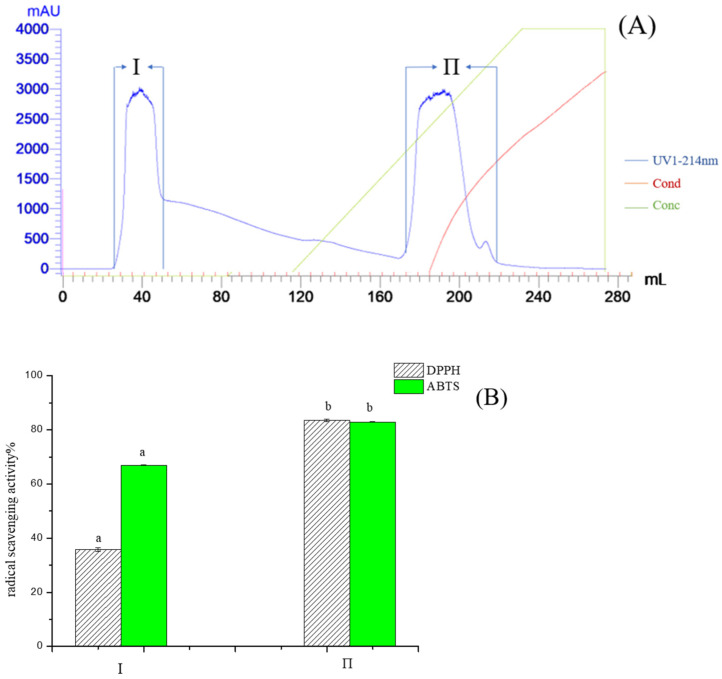
Elution profile of PPH-I on Q Sepharose Fast Flow column (1.6 × 30 cm), and activities of each fraction. (**A**) Chromatogram profiles of PPH-I; (**B**) DPPH and ABTS radical scavenging activities of fraction I and fraction II. Different letters in the same test indicate significant differences (*p* < 0.05).

**Figure 3 molecules-28-02952-f003:**
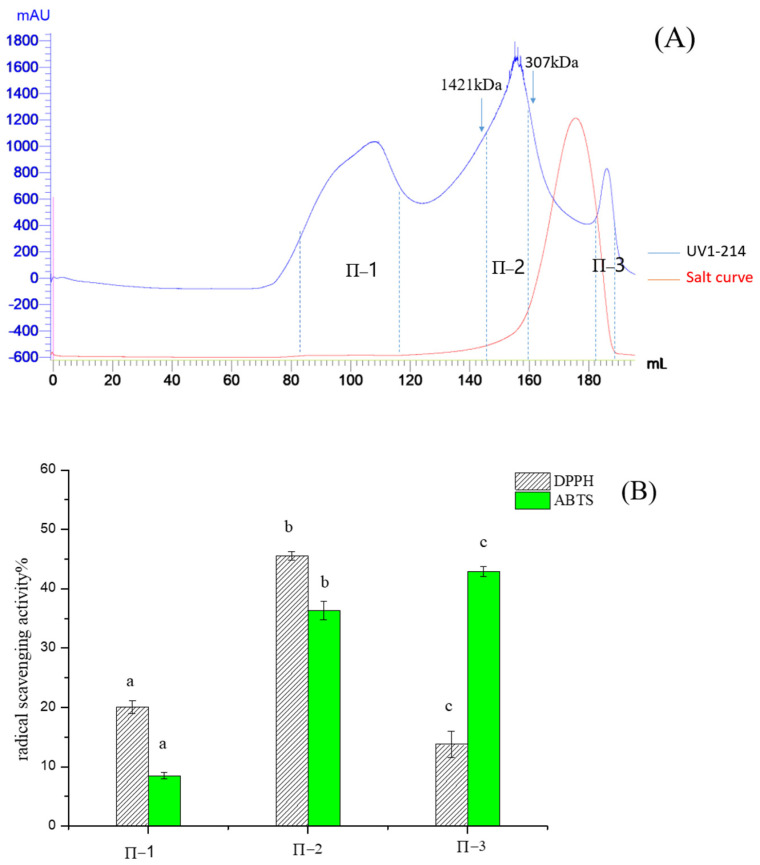
Elution profile of fraction II loaded on Sephadex G-25 column (1.6 × 90 cm) and activities of each fraction. (**A**) Chromatogram profiles of fraction II; (**B**) DPPH and ABTS radical scavenging activities of fraction II-1, fraction II-2 and fraction II-3. Different letters in the same test in (**B**) indicate significant differences (*p* < 0.05).

**Figure 4 molecules-28-02952-f004:**
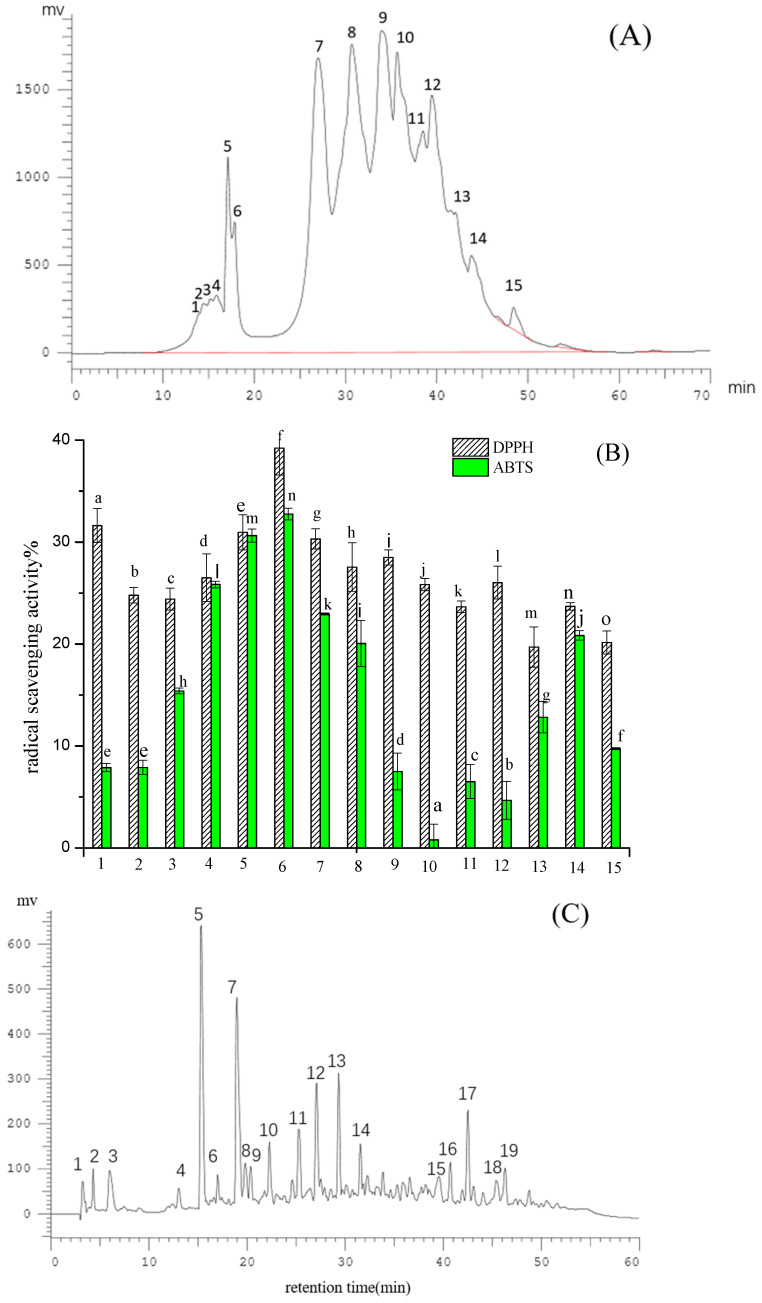
Reversed-phase chromatogram fraction II-2 and activities of each fraction: elution profile of fraction II-2 by ProntoSIL C18 ace-EPS column (**A**), DPPH and ABTS radical scavenging activities of 15 eluting fractions (**B**); elution profile of fraction II-2-6 by the analytical Cosmosil pbr column (Ø4.6 × 250 mm, 5 μm) (**C**), and DPPH and ABTS radical scavenging activities of 19 eluting fractions (**D**). different letters in the same test in (**B**,**D**) indicate significant differences (*p* < 0.05).

**Figure 5 molecules-28-02952-f005:**
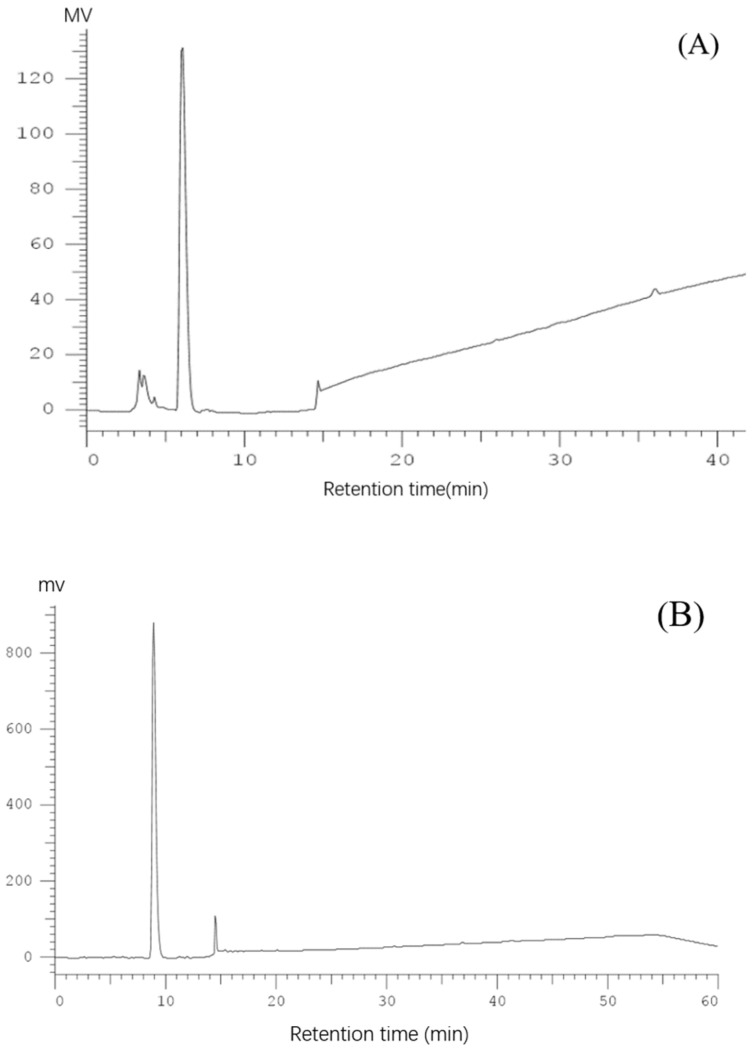
Elution profile of fraction loaded on Xselect TM CSH130 (Ø4.6 × 250 mm, 3.5 μm). Elution profile of fraction II-2-6-3 (**A**); elution profile of fraction II-2-6-5 (**B**); elution profile of fraction II-2-6-7 (**C**).

**Figure 6 molecules-28-02952-f006:**
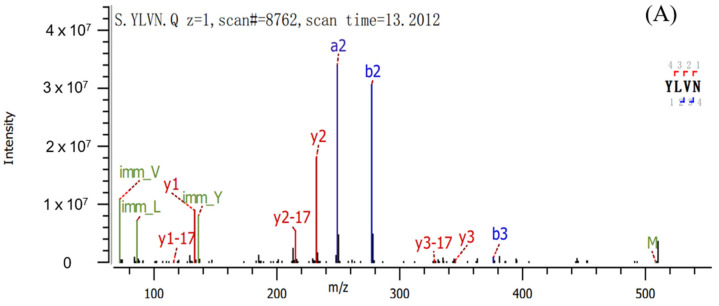
LC-MS/MS analysis of the active fractions. (**A**) FII-2-6-3; (**B**) FII-2-6-5; (**C**) FII-2-6-7.

**Figure 7 molecules-28-02952-f007:**
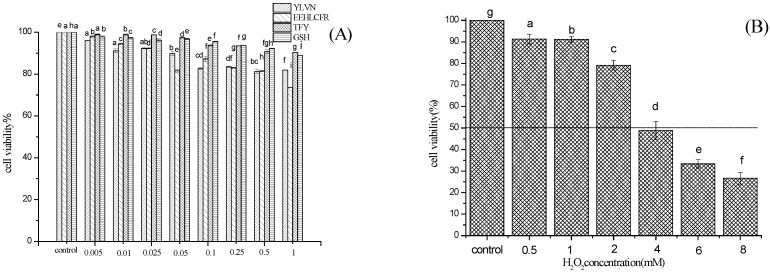
Effects of three peptides in H_2_O_2_-injured LO2 cells oxidative stress: Cell viability (**A**) Establishment of hydrogen peroxide concentration (**B**), Pre-protective effect of peptides (**C**). Different letters indicate significant differences (*p* < 0.05). ### *p* < 0.001 versus the blank control group; ** *p* < 0.01 versus the H_2_O_2_-induced group.

**Figure 8 molecules-28-02952-f008:**
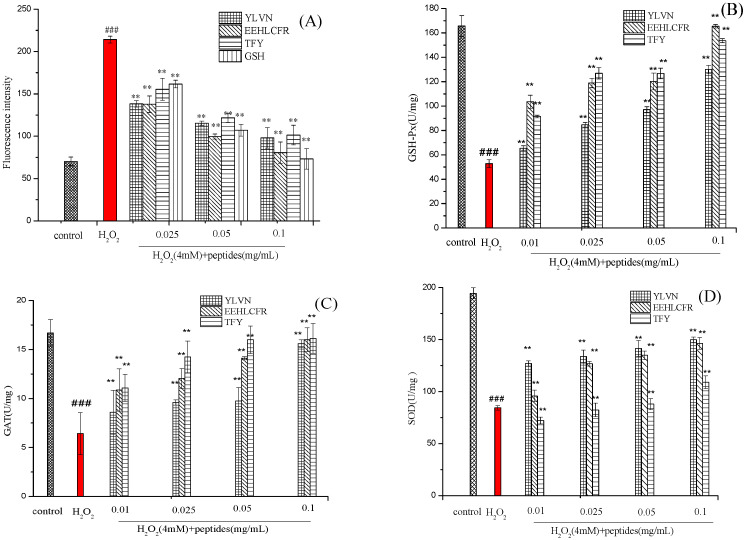
Effects of different peptide concentration treatments on H_2_O_2_-stressed cells. The ROS levels (**A**), GSH-Px (**B**), CAT (**C**) and SOD (**D**). ### *p* < 0.001 versus the control group; ** *p* < 0.01 versus the H_2_O_2_-induced group.

**Figure 9 molecules-28-02952-f009:**
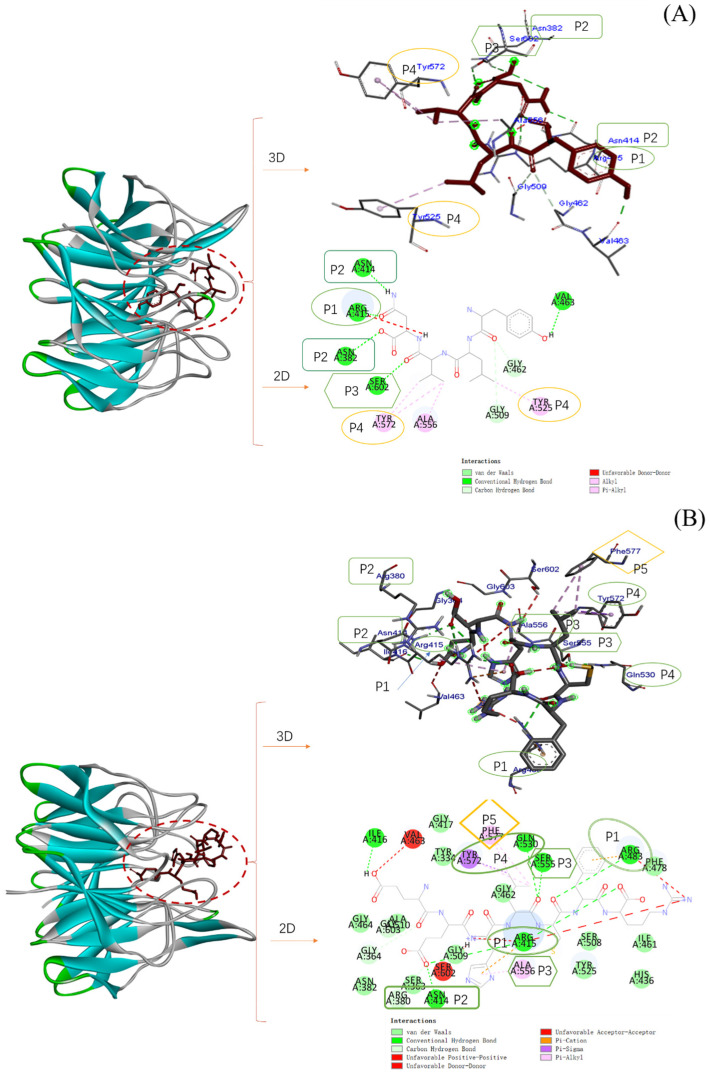
The 2D and 3D interactions between Keap1 (4IQK) and pea antioxidant peptides. (**A**) Peptide 1 YLVN. (**B**) Peptide 2 EEHLCFR, and (**C**) Peptide 3 TFY.

**Table 1 molecules-28-02952-t001:** Sequence, length, molecular mass and the hydrophobic ratio of three novel antioxidant peptides.

Peptide	Sequence	Length	Mass	Hydrophobic Ratio ^a^ (%)
Peptide 1	YLVN	4	507.59	50.00
Peptide 2	EEHLCFR	7	933.06	28.57
Peptide 3	TFY	3	429.47	33.33

^a^ Hydrophobic residues in the peptide sequence include A, V, G, L, P, I, M, F, and W.

**Table 2 molecules-28-02952-t002:** Antioxidant activity of three novel antioxidant peptides in vitro.

Amino Acid Sequence	DPPH· IC_50_ (mg/mL)	OH· IC_50_ (mg/mL)	ABTS+ IC_50_ (mg/mL)	O_2_^−^· IC_50_ (mg/mL)	ORAC (μmol TE/μmol Peptide)
YLVN	-	-	0.002 ± 0.000 ^a^	1.357 ± 0.458 ^c^	1.120 ± 0.231 ^a^
EEHLCFR	0.027 ± 0.003 ^c^	2.796 ± 0.597 ^a^	0.019 ± 0.001 ^b^	1.247 ± 0.041 ^b^	0.921 ± 0.131 ^b^
TFY	1.492 ± 0.001 ^b^	-	0.006 ± 0.004 ^ab^	-	0.367 ± 0.092 ^c^
GSH	0.081 ± 0.015 ^a^	0.102 ± 0.091 ^b^	0.007 ± 0.003 ^ab^	0.667 ± 0.009 ^a^	0.484 ± 0.190 ^d^

The sample concentration is set in the range of 0–3 mg/mL, - which means that no IC_50_ value has been calculated at this concentration. The ORAC antioxidant activity of the synthetic peptide was determined at 0.5 mg/mL. Different letters superscript on the same column indicate significant differences (*p* > 0.05).

**Table 3 molecules-28-02952-t003:** Keap1 (4IQK) docking results and interaction sites with pea antioxidant peptides.

Peptide	Affinity (kcal/mol)	Hydrogen Bond	Hydrophobic	Electrostatic
YLVN	−8.2	Asn414, Arg415, Asn382, Ser602, Val463, Gly462, Gly509	Tyr572, Tyr525, Ala556	
EEHLCFR	−7.2	Arg415, Arg483, Ser555, Gln530, Ile416, Asn414, Arg380, Gly364, Gly603	Phe577, Ala556	Tyr572
TFY	−8.9	Ser508, Arg415, Asn414, Ser363, Ala556	Tyr334, Ala556	

## Data Availability

All the data generated by this research are included in the article.

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
