# Peer review of "Purification, Identification and Evaluation of Antioxidant Peptides from Pea Protein Hydrolysates"

_molecules, 2023, doi:10.3390/molecules28072952_

Round 1

Reviewer 1 Report

In this manuscript the purification and identification of antioxidant peptides from a pea protein enzymatic hydrolysate is described. This is an interesting and comprehensive study where a  wide range of methodology has been applied, not only for the identification of the peptides but also to obtain a better insight into the activity by the structure-activity relationship analysis and molecular docking approach.  The biological antioxidant activity is also investigated further by measuring the protective effect of peptides against H2O2 induced oxidative stress in LO2 cells.

The manuscript is generally well written however, there is some key info missing in some of the methods that needs to be added. See the recommendations below.

Line 49. Authors say total content, is this dry weight?

Explain what LO2 cells are, what type of cells?

Line 72. Section 2.2. Pea protein is supplied by?  Or is it produced by the authors?

Hydrolysis. Missing enzyme to substrate ratio

Line 153- Statistics. Explain what do you mean by “All trials were measured in 3 replicates”

Explain if hydrolysis were carried out in triplicate-and analysis were carried out in triplicate

Method 2.3- Authors should refer to Fig 1 where they describe clearly the purification process. Missing to add another  RP-HPLC step with a different column in the description as in Fig 1 and in results. Also missing sample volume.

Method 2.6. Antioxidant activity- Authors should include briefly a description not just a reference to the method. It is important to show that enough sample and volume are produced particularly, after the RP-HPLC steps , last two steps of purification to carry out the antioxidant tests.

Reviewer 2 Report

The manuscript “Purification, identification and evaluation of antioxidant peptides from pea protein hydrolysates” has scientific importance in the subject areas of the journal. However, some deficiencies should be improved before the possible publication of the manuscript.

Lines 15-16 and subsequent. Insert a space to separate mg/mL units.

Line 52. Include the reference of it previous study.

Line 72. How was pea protein obtained? Please explain.

Lines 72-74. Is this method previously reported? If so, please include reference.

Line 96 and subsequent. Insert a space to separate μL units.

Line 106 and subsequent. Reference style should be similar. For in-text citations, I suggest put last name (without initials) and include et al. (period at the end) for a work with three or more authors.

Line 107. Lower case in the word The.

Line 135. Delete the hyphen (24-h).

Line 148. Insert a space before a (chemaxon.com)a

Line 149. Change the tense is performed by was performed.

Line 150. Change pymol by Pymol (capital letter).

Lines 160-161. The authors mentioned that PPHI, PPHII, and PPHIII fractions not showed significant differences in the free radical scavenging (by DPPH and ABTS). What were the values determined?

Line 163. In the figure 1 only is observed one step in which activity is tracked, denoted as the best active fraction. So, why authors mention that the activities (DPPH and ABTS) of fractions from each purification step were tracked?

Line 164. Change Figure 1 by Fig 1.

Line 167. Please include a description of legends at the bottom of the figure.

Line 172. Change PH7.5 by pH 7.5

Line 203 and subsequent. Insert a space before (A-C).

Lines 211-215, Fig 4 description. Include space before each (), change Elution word by elution (lower case).

Line 263. Change antioxidant word by Antioxidant.

Be consistent in denoting the p-value, I suggest to use lower case throughout the text.

Line 391. Insert a space before YLVN peptide.

Line 409. Change Table3 by Table 3. Change Kcal/mol by kcal/mol.

Line 410. Are P1, P2, P3, P4 and P5 subregions mentioned in Fig. 9 the pockets described in lines 413 to 420?

In 3-D models (Fig 9), please show P1, P2, P3, P4 and P5 subregions or pockets.

Insert period at the end of all table titles.

Check the writing of references according to the instructions for authors

I suggest review that the name of peptides to be similar in all document because in some lines appear as Peptide 1 (line 384), peptide1 (line 273) or peptide 1 (line 282).

Line 433. Chane Fig.9. by Fig 9.
